# Shelf-Supervised Cross-Modal Pre-Training for 3D Object Detection

**Mehar Khurana**[1][*]**, Neehar Peri**[2*]**, James Hays**[3]**, Deva Ramanan**[2]
[1]IIIT Delhi, [2]Carnegie Mellon University, [3]Georgia Institute of Technology

**Abstract:** State-of-the-art 3D object detectors are often trained on massive labeled datasets. However, annotating 3D bounding boxes remains prohibitively expensive and time-consuming, particularly for LiDAR. Instead, recent works demonstrate that self-supervised pre-training with unlabeled data can improve detection accuracy with limited labels. Contemporary methods adapt best-practices for self-supervised learning from the image domain to point clouds (such as contrastive learning). However, publicly available 3D datasets are considerably smaller and less diverse than those used for image-based self-supervised learning, limiting their effectiveness. We do note, however, that such 3D data is naturally collected in a multimodal fashion, often paired with images. Rather than pre-training with only self-supervised objectives, we argue that it is better to bootstrap point cloud representations using image-based foundation models trained on internet-scale data. Specifically, we propose a *shelf*-supervised approach (e.g. supervised with off-the-shelf image foundation models) for generating zero-shot 3D bounding boxes from paired RGB and LiDAR data. Pre-training 3D detectors with such pseudo-labels yields significantly better semi-supervised detection accuracy than prior self-supervised pretext tasks. Importantly, we show that image-based shelf-supervision is helpful for training LiDAR-only, RGB-only and multi-modal (RGB + LiDAR) detectors. We demonstrate the effectiveness of our approach on nuScenes and WOD, significantly improving over prior work in limited data settings. Our code is available on GitHub.

**Keywords:** Shelf-Supervised 3D Object Detection, Semi-Supervised Learning, Vision-Language Models, Autonomous Vehicles

## 1 Introduction

3D object detection is an integral component of the robot perception stack. To facilitate research in this space, the Autonomous Vehicle (AV) industry has released several large-scale 3D annotated multi-modal datasets [1, 2, 3]. However, 3D detectors pre-trained on nuScenes [1] cannot be easily applied to Argoverse [3] due to differences in sensor characteristics across hardware platforms. In practice, training 3D detectors for a new hardware platform requires re-annotating 3D bounding boxes at scale, which can be prohibitively expensive [4] and time consuming [5]. Instead, recent works [6, 7, 8, 9] demonstrate that self-supervised pre-training with unlabeled sensor data can improve downstream detection accuracy with limited labels. Notably, AVs *already* capture large-scale unlabeled multi-modal data localized to HD maps during normal testing [10], motivating the exploration of self-supervised pre-training for outdoor scenes.

**Status Quo.** Self-supervised learning offers a scalable framework to learn from massive unlabeled datasets. Typically, self-supervised methods establish a pretext task that derives supervision directly from the data (e.g. occupancy prediction, contrastive learning, or masked auto-encoding). Once trained, these self-supervised representations can be adapted for downstream tasks by fine-tuning on

---

[*]Equal Contribution.

8th Conference on Robot Learning (CoRL 2024), Munich, Germany.

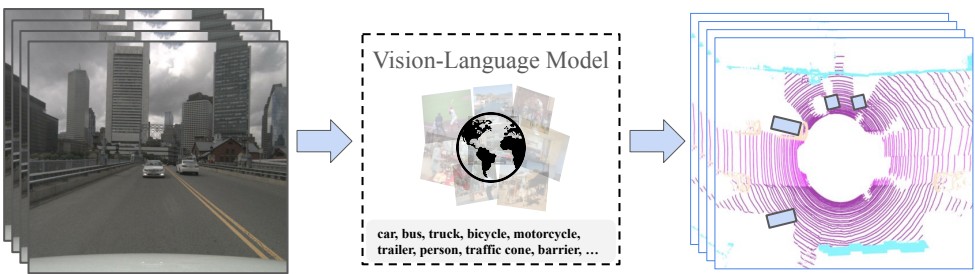

Figure 1: **Cross-Modal 3D Detection Distillation with Vision-Language Models.** Existing datasets used for 3D representation learning are considerably smaller and less diverse than those used for image-based self-supervised learning, limiting the effectiveness of pre-training. Therefore, we propose a simple approach for transferring object-centric priors from vision-language models to LiDAR. Specifically, we project 3D LiDAR points onto 2D instance segmentation masks to generate zero-shot 3D bounding boxes [10] that can be used for pre-training 3D detectors.

a limited set of annotations. Given the success of image-based self-supervised learning [11, 12, 13], recent approaches revisit contrastive learning in the context of 3D sensor data [9, 14]. However, existing 3D datasets are considerably smaller and less diverse than those used in image-based self-supervised learning. Instead of solely relying on self-supervised objectives for pre-training, we advocate for a more practical approach that embraces recent advances in image-based foundational models by bootstrapping point cloud representations with vision-language models (VLMs) trained on internet-scale data. Since off-the-shelf vision-language models *already* encode semantics and object priors, our work focuses on distilling these 2D foundational priors into 3D detectors.

**Bootstrapping 3D Representations with 2D Priors.** VLMs [15, 16, 17] are trained on internet-scale data and show impressive zero-shot detection performance across many application domains [18]. Therefore, we posit that pre-training with noisy pseudo-labels distilled from vision-language models can outperform contrastive features learned from scratch (cf. Fig. 1). However, distilling image-based foundational knowledge to LiDAR remains an open challenge [19, 20, 21]. We address this challenge by explicitly "inflating" 2D instance masks to 3D bounding boxes via unlabeled multimodal (RGB + LiDAR) training data, using a careful combination of LiDAR cues, HD maps, and shape priors [10]. This allows us to produce accurate 3D pseudo-labels which can then be used to pre-train uni-modal or multi-modal 3D detectors. One significant advantage of our *shelf*-supervision over self-supervision is that it naturally produces a more aligned pretext task; instead of pre-training 3D detectors with contrastive learning, we learn from 3D bounding box pseudo-labels.

**Contributions**. We present three major contributions. First, we propose Cross-Modal 3D Detection Distillation (CM3D), a zero-shot approach for generating 3D bounding boxes using off-the-shelf vision-language models. Notably, we show that pre-training detectors with 3D bounding box pseudo-labels achieves higher downstream detection accuracy than prior contrastive learning objectives. We conduct extensive experiments to ablate our design choices and demonstrate that our simple approach achieves state-of-the-art unsupervised and semi-supervised 3D detection accuracy on the nuScenes and WOD benchmarks.

## 2 Related Works

**Unsupervised 3D Object Detection** has gained significant interest in recent years as a way to auto-label large datasets without human annotators. Dewan et al. [22] introduced a model-free approach to detect and track the visible portions of objects by leveraging motion cues from LiDAR. More recently, Wong et al. [23] identified unknown instances through supervised segmentation and clustering [24]. However, both [22, 23] only reason about the visible extent of objects and are unable to generate amodal bounding boxes. Cen et al. [25] used a supervised detector to generate bounding

box proposals for unknown categories, and Tian et al. [26] exploited the correspondence between images and point clouds to generate object proposals. Similarly, Wilson et al. [10] employed an HD map and shape priors to "inflate" 2D detections to 3D cuboids. Utilizing priors such as surface normals [27, 28, 29] and motion cues [30, 31, 32, 33] has been shown to improve unsupervised object detection. You et al. [34] integrated cues from temporal scene changes (e.g. point ephemerality from multiple traversals) to detect mobile objects [35, 36]. Najibi et al. [37] used scene flow and clustering to generate 3D bounding boxes for moving objects. However, this method does not assign semantic labels to cuboids and cannot detect static objects. Similarly, [38, 39] use scene flow to cluster points and generate pseudo-labels.

**Self-Supervised Learning for 3D Representations** has been applied to many domains, including object-centric point clouds [40, 41, 42], indoor scenes [43, 44, 45, 46, 47, 48], and outdoor environments [9, 7, 49]. Current techniques for self-supervised point cloud pre-training can be broadly characterized by their receptive field. Scene-based contrastive learning [6, 50] (often used in the context of representation learning for indoor scenes) may not capture the necessary fine-grained details required for recognizing small objects. In contrast, voxel-based contrastive learning (often used in the context of representation learning for outdoor scenes) inherently struggles with a restricted receptive field, limiting its ability to encode larger structures. Region-based representations [7, 51] attempt to strike a balance between coarse-grained scene representations and fine-grained voxel representations. However, most regions in outdoor scenes (e.g. ground plane and buildings) lack informative cues, making it difficult to learn robust features. More recently, multi-modal contrastive learning methods [8, 9] demonstrated that paired multi-modal sensors can provide complementary (self-)supervision for pre-training. PointContrast [6] established correspondences between different views of point clouds using a contrastive loss. DepthContrast [14] treated different depth maps as contrastive instances and discriminates between them. STRL [52] extracted invariant representations from temporally correlated frames in a 3D point cloud sequence. More recently, SimIPU [8] and CALICO [9] delve into multi-modal contrastive learning by utilizing paired RGB and LiDAR data. GeoMAE [49] frames point-cloud representation learning as a masked auto-encoding task, and uses geometric pretext tasks including occupancy, normals, and curvature estimation.

**Leveraging 2D Foundational Models for 3D Perception** is an active area of research. Sautier et al. [53] introduced SLidR, a 2D-to-3D representation distillation method aimed at cross-modal self-supervised learning on large-scale point clouds. Mahmoud et al. [54] subsequently extended SLidR with a semantically-aware contrastive constraint and a class-balancing loss. SEAL [55] takes inspiration from SLidR and used SAM [56] to generate 3D class-agnostic point cloud segments for contrastive learning. More recently, SA3D [57] extends SAM [56] to segment 3D objects with NeRFs. Anything-3D [58] combines BLIP [16], a pre-trained 2D text-to-image diffusion model, with SAM for single-view conditioned 3D reconstruction. 3D-Box-Segment-Anything utilizes SAM with a pre-trained VoxelNeXt [59] 3D detector for interactive labeling. SAM3D [60] repurposes SAM to directly segment objects from BEV point clouds. Most recently, UP-VL [20] distilled 2D vision-language features [11] into LiDAR point clouds to generate amodal cuboids.

## 3 Cross-Modal 3D Detection Distillation (CM3D) with VLMs

Given a large unlabelled set of multi-modal (RGB + LiDAR) data, we combine 2D VLMs with 3D domain knowledge (given by 3D shape priors and map constraints) to generate 3D bounding box pseudo-labels for pre-training 3D detectors (Fig. 2). Using these pseudo-labels, we can train LiDAR-only, RGB-only, or mulit-modal 3D detectors.

### 3.1 Generating 3D Bounding Box Pseudo-Labels

**2D Mask Generation.** The first stage of our pipeline requires producing accurate 2D instance masks for a fixed vocabulary of object categories. We prompt a 2D VLM detector (e.g., Detic [15] or GroundingDINO [61]) with class names (e.g., `car`, `bus`, `truck`) to generate 2D box proposals. Although some VLM detectors (i.e., Detic) already produce instance segmentation masks, we found

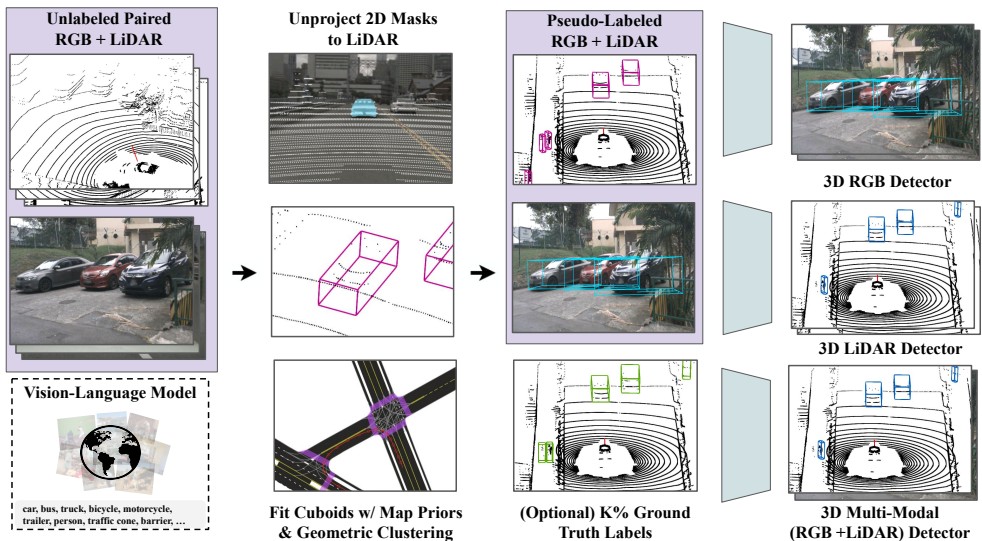

Figure 2: **Overview.** Given unlabeled, calibrated, and paired RGB images and LiDAR sweeps, we generate 3D bounding boxes psuedo-labels using foundational 2D priors from VLMs, map priors, and geometric clustering. We describe our unprojection step further in Fig. 3. Using these pseudo-labels, we can train LiDAR-only, RGB-only, or multi-modal 3D detectors.

it more effective to prompt separate (foundation) models such as SAM [56] with the predicted 2D bounding boxes to generate high-quality segmentation masks. We evaluate the impact of using different VLMs in Appendix F. Notably, we find that Detic [15] produces better 3D pseudo-labels than GroundingDINO [61] in our CM3D pipeline.

**3D Projection.** Next, we project the LiDAR points onto the 2D image plane and group all points within each instance mask. To produce a final 3D bounding box pseudo-label, we need to generate a 3D centroid, 3D orientation, and cuboid dimensions (width, length, height). Our key technical approach, described below and inspired by [10], is to leverage 3D priors in the form of maps (e.g., car cuboids should typically be aligned with map geometry) and shape priors (e.g., fine-grained categories such as trucks tend to have canonical dimensions).

**3D Cuboid Generation.** We define an initial candidate 3D centroid to be the medoid of points within each mask, expressed as $m = \operatorname{argmin}_{x \in P_M} \sum_{p \in P_M} ||p - x||_2$, where $P_M$ is the set of Li-DAR points that lie within the mask $M$. Note that this medoid tends to lie on the surface of the object visible to the LiDAR sensor rather than the true object centroid, which implies it will require refinement. To estimate a canonical width-length-height, we simply prompt an LLM (e.g., Chat-GPT) with each class name and use the returned value for all instances. Although LLMs do not have access to our 3D training data, they have seen many descriptions of object shapes on the web and can provide reasonable prototypical 3D sizes of common objects (Appendix H). Lastly, we estimate box orientation for vehicles using lane geometry from an HD map and assign the direction of the nearest lane to each cuboid. We assign an arbitrary orientation to all non-vehicles (e.g., `pedestrians`, `barriers` and `traffic cones`).

**3D Refinement.** Fig. 3 visualizes our overall pseudo-label generation pipeline. Because many components of our pipeline provide only rough 3D estimates (e.g., not all trucks have precisely the same 3D dimension), we find that 3D labels can be refined. Table 1 ablates several strategies for improvement, including prompt engineering to improve VLM zero-shot 2D detections, LiDAR accumulation to improve medoid estimation, mask erosion to remove noisy LiDAR points near mask boundaries, medoid compensation to reduce object center estimation biases, and late-fusion of independent zero-shot 3D detectors to improve orientation and shape estimation. We find that prompt engineering and medoid compensation provide the greatest improvement to pseudo-label quality. We present a detailed overview of each refinement step in Appendix B.

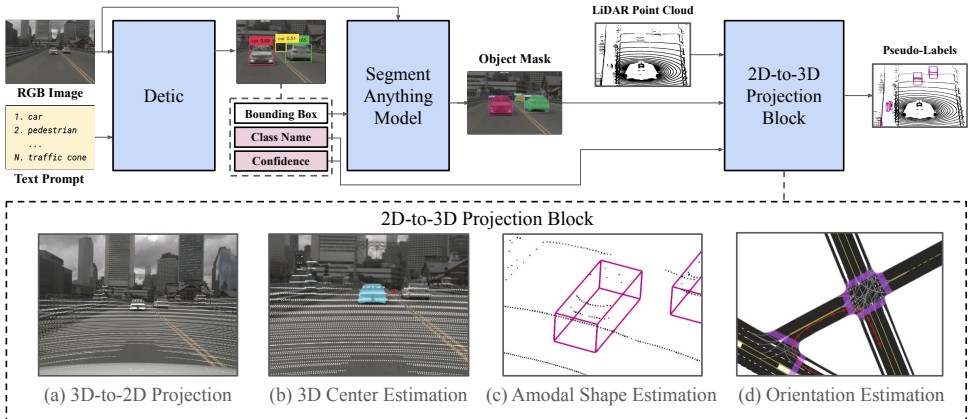

Figure 3: **Unprojecting 2D Foundational Priors to 3D.** First, we prompt an open-vocabulary 2D detector (e.g., Detic [15]) with a class name (e.g., `car`) to generate 2D box proposals. Next, we prompt SAM [56] with the predicted 2D bounding boxes to generate high-quality instance segmentation masks. We then generate an oriented 3D cuboid using the set of LiDAR points that project to a given 2D instance mask. Specifically, we define the *center* of the cuboid to be the medoid of the LiDAR points, the *dimensions* (length, width, height) to be a fixed shape prior (similar to an anchor box) as reported by ChatGPT when prompted with the class name, and the *orientation* to be aligned with lane geometry provided from an HD map.

## 3.2 Training with Pseudo-Labels

We use our generated pseudo-labels to pre-train 3D detectors that will be fine-tuned on a limited set of labeled data. We present a block diagram of our training pipeline in Fig 2. Notably, it is trivial to train *any* 3D detector with our pseudo-labels, making our proposed framework widely applicable. Further, although our pseudo-label generator requires paired RGB and LiDAR data for pseudo-label generation, we can easily train a LiDAR-only model like CenterPoint, which requires only LiDAR sweeps at inference. In this paper, we evaluate CM3D by training two popular detectors, CenterPoint [62] and BEVFusion [63]. Following CALICO's [9] experiment setup, we train BEVFusion with the CenterHead and turn off class-balanced grouping and sampling. Although copy-paste augmentation is widely used when training 3D detectors, we find that performance degrades with our noisy pseudo-labels (Appendix E). We train the CenterHead following standard 3D detection losses. We use the sigmoid focal loss (for recognition) [64] and L1 regression loss (for localization). Concretely, our loss function is as follows: $L = L_{HM} + \lambda L_{REG}$, where $L_{HM} = \sum_{i=0}^{C} SigmoidFocalLoss(X_i, Y_i)$ and $L_{REG} = |X_{BOX} - Y_{BOX}|$, where $X_i$ and $Y_i$ are the $i^{th}$ class' predicted and ground-truth heat maps, while $X_{BOX}$ and $Y_{BOX}$ are the predicted and pseudo-label box attributes.

**Self Training.** Although directly pre-training with our pseudo-labels significantly improves mAP, it also yields lower NDS compared to prior work. We posit that this is due to the errors in size and orientation estimation, which can both be attributed to our simplistic 3D priors. Prior work suggests that self-training can be effective for semi-supervised learning [34, 20]; not only can it discover more objects [34, 20], but it may also refine noisy box labels. We ablate the impact of self-training in the Appendix D. We find that even one round of self-training significantly improves average precision and reduces orientation estimation error.

## 4 Experiments

In this section, we conduct extensive experiments to evaluate our proposed approach. We find that CM3D achieves state-of-the-art unsupervised detection performance and significantly improves over

Table 1: **Pseudo-Label Refinement**. We analyze the impact of each component over the baseline [10] (cf. Fig 3). Importantly, we find that prompt engineering, medoid compensation, and non-maximum suppression have the greatest impact. See Appendix B for additional details.

| Method | mAP ↑ | NDS ↑ |
|---|---|---|
| Baseline (Re-implementation of [10]) | 18.6 | 17.8 |
| + Prompt Engineering | 19.7 (+1.1) | 18.4 (+0.6) |
| + LiDAR Accumulation | 20.0 (+0.3) | 18.6 (+0.2) |
| + Mask Erosion | 20.2 (+0.2) | 19.1 (+0.5) |
| + Medoid Compensation | 21.8 (+1.6) | 21.3 (+2.1) |
| + Non-Maximal Suppression | 22.8 (+1.0) | 21.9 (+0.6) |
| + Late Fusion | 23.0 (+0.2) | 22.1 (+0.2) |

prior works in limited data settings (cf. Table 3). We copy the results of prior work from CALICO and refer readers to [9] for further details.

**Datasets**. We evaluate our work using the well-established nuScenes dataset [1]. We follow the suggested protocol in [7, 49, 9] and sample K% of the training data uniformly from the entire training set. Lastly, we also evaluate our approach on the Waymo Open Dataset [2] in Appendix I, and find that our conclusions hold across datasets.

**Metrics**. Mean average precision (mAP) is an established metric for object detection [65]. For 3D detection, a true positive (TP) is defined as a detection that has a center distance within a distance threshold on the ground-plane to a ground-truth annotation [1]. mAP computes the mean of AP over all classes, where per-class AP is the area under the precision-recall curve drawn with distance thresholds of [0.5, 1, 2, 4] meters. NDS (nuScenes Detection Score) is computed by taking a weighted sum of mAP along with five other metrics, including mATE (mean absolute translation error), mAOE (mean absolute orientation error), mASE (mean absolute scale error), mAVE (mean absolute velocity error), and mAAE (mean absolute attribute error). mAP is assigned a weight of 5, while all other metrics are assigned a weight of 1.

**Pseudo-Label Refinement.** We ablate our pseudo-label generation algorithm to determine how each component improves the baseline. As discussed above, we find that medoid compensation has the greatest impact on pseudo-label quality, improving mAP by 1.6% and NDS by 2.1%. Tuning the Detic text prompts by including synonyms also improves results significantly. Finally, the 3D distance-based NMS helps remove duplicates present in the overlapping regions of the ring cameras and increases the mAP by 1%.

**Class Agnostic Evaluation.** Prior pseudo-label generation methods only evaluate class-agnostic performance [39]. For fair comparison with prior work, we evaluate CM3D pseudo-labels without differentiating between classes in Table 2. Prior works use motion cues from scene flow to generate

Table 2: **Class Agnostic Evaluation on nuScenes**. We evaluate class-agnostic performance of CM3D pseudo-labels for fair comparison with prior work. We find that our method significantly outperforms LISO-TF by 14.5% AP.

| Method | Modality | AP ↑ | NDS ↑ | mATE ↓ | mAOE ↓ | mASE ↓ |
|---|---|---|---|---|---|---|
| DBSCAN [24] | L | 0.8 | 10.9 | 0.980 | 3.120 | 0.962 |
| RSF [66] | L | 1.9 | 18.3 | 0.774 | 1.003 | 0.507 |
| Oyster-CP [67] | L | 9.1 | 21.5 | 0.784 | 1.514 | 0.521 |
| Oyster-TF [67] | L | 9.3 | 23.3 | 0.708 | 1.564 | 0.448 |
| LISO-CP [39] | L | 10.9 | 22.4 | 0.750 | 1.062 | 0.409 |
| LISO-TF [39] | L | 13.4 | 27.0 | 0.628 | 0.938 | **0.408** |
| CM3D (Ours) | L + C | **27.9** | **27.6** | **0.592** | **0.872** | 0.428 |

Table 3: **Semi-Supervised 3D Detection on nuScenes**. In terms of zero-shot accuracy, pseudo-labels from CM3D outperforms prior art (SAM3D) by 20.7% NDS. With 5% supervised training data, pre-training CenterPoint with CM3D pseudo-labels improves over the prior art of PRC [9] by 8.1 mAP / 2.8 NDS. When comparing to Camera-LiDAR methods, pre-training BEVFusion with CM3D pseudo-labels outperforms prior art (CALICO) by 8.6 mAP / 4.6 NDS. We highlight the best LiDAR-only (L) results in blue, the best Camera-only (C) results in green, and the best Camera-LiDAR (L+C) results in red.

| Training Data | Method | Modality | | mAP ↑ | NDS ↑ |
| --- | --- | --- | --- | --- | --- |
| | | Train | Test | | |
| 0%
(Unsupervised) | SAM3D [60] | L | L | 1.7 | 2.4 |
| | CM3D (Ours) | L + C | L + C | 23.0 | 22.1 |
| | CenterPoint [62] + CM3D (Ours) | L + C | L | 16.7 | 21.4 |
| | BEVFusion-C [63] + CM3D (Ours) | L + C | C | 11.7 | 16.1 |
| | BEVFusion [63] + CM3D (Ours) | L + C | L + C | 20.6 | 23.3 |
| 5% | CenterPoint [62] (Rand. Init.) | L | L | 33.1 | 37.4 |
| | PointContrast [6] | L | L | 36.7 | 43.0 |
| | ProposalContrast [7] | L | L | 37.0 | 43.1 |
| | PRC [9] | L | L | 38.2 | 46.0 |
| | CenterPoint [62] + CM3D (Ours) | L + C | L | 46.3 | 48.8 |
| | BEVFusion [63] (Rand. Init.) | L + C | L + C | 39.0 | 43.7 |
| | SimIPU [8] | L + C | L + C | 39.1 | 45.8 |
| | PRC [9] + BEVDistill [68] | L + C | L + C | 41.0 | 47.5 |
| | CALICO [9] | L + C | L + C | 41.7 | 47.9 |
| | BEVFusion [63] + CM3D (Ours) | L + C | L + C | 51.3 | 52.5 |
| 10% | CenterPoint [62] (Rand. Init.) | L | L | 41.1 | 48.0 |
| | PointContrast [6] | L | L | 42.3 | 51.2 |
| | ProposalContrast [7] | L | L | 42.1 | 51.1 |
| | PRC [9] | L | L | 44.1 | 53.1 |
| | CenterPoint [62] + CM3D (Ours) | L + C | L | 51.0 | 56.3 |
| | BEVFusion [63] (Rand. Init.) | L + C | L + C | 46.2 | 51.9 |
| | SimIPU [8] | L + C | L + C | 47.5 | 52.4 |
| | PRC [9] + BEVDistill [68] | L + C | L + C | 49.7 | 53.6 |
| | CALICO [9] | L + C | L + C | 50.0 | 53.9 |
| | BEVFusion [63] + CM3D (Ours) | L + C | L + C | 53.3 | 56.5 |

object proposals for moving objects [38]. In contrast, our approach localizes both static and moving objects and classifies them using VLMs. CM3D outperforms LISO-TF [39] by 14.5% AP and 0.6% NDS, highlighting the benefit of using foundational priors and multi-modal inputs.

**Evaluating Zero-Shot Performance**. We evaluate the quality of CM3D's pseudo-labels (0% Training Data) and find that our method achieves 22.3% mAP and 22.1% NDS. We compare CM3D with SAM3D [60], a recently released zero-shot 3D detector. Notably, both CM3D and SAM3D use SAM [56] to group LIDAR points. However, CM3D groups LiDAR points on the image plane and SAM3D [60] groups LiDAR points in the 2D BEV plane. Our multi-modal zero-shot 3D detector significantly outperforms SAM3D's LiDAR-only approach by 20.7%. Importantly, SAM3D was initially designed as a class-agnostic zero-shot detector. We posit that SAM3D's poor performance can be attributed to nuScenes' diverse classes.

**Distilling Multi-Modal Priors into Uni-Modal Models.** Although CM3D requires paired RGB images and LiDAR sweeps for pseudo-label generation, we can easily train an RGB-only or LiDAR-only student model (e.g., CenterPoint + CM3D). Importantly, these student models don't require paired RGB-LiDAR data at inference (and are therefore compared against other models that only use LiDAR at inference). However, we find that the CenterPoint + CM3D (0% Training Data) student model performs worse than the pseudo-label generator (CM3D). This can be attributed to learning with noisy labels and not having access to the same sensor data as the teacher. Interestingly, we find that using multi-modal cues for both training and testing yields higher results.

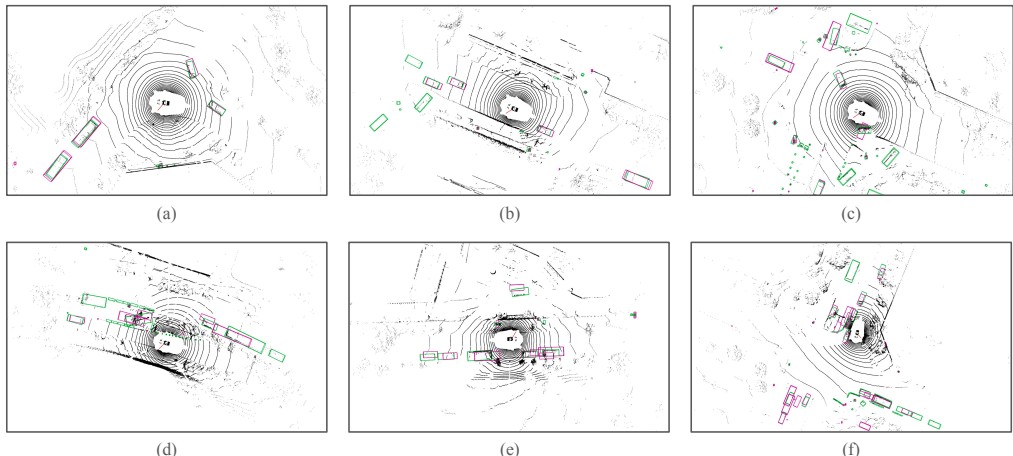

Figure 4: **Qualitative Results of Pseudo-Labels**. We visualize pseudo-labels (pink) and ground-truth labels (green) across all 10 object classes on the nuScenes val-set. In (**a**), our pseudo-labels accurately estimate location, cuboid size, and orientation, demonstrating the general effectiveness of medoid compensation and map-based orientation estimation. In (**b**), we find that CM3D often misses heavily-occluded objects. This is unsurprising because our method relies on accurate RGB-based detections, which often fail with heavy occlusions. In (**c**), our map-based orientation estimation fails when the predicted object is not oriented in the direction of any lane. For example, the incorrect orientation of the `car` turning into the intersection (not aligned to any nearby lanes) illustrates the limitations of our approach. In both (**d**) and (**f**), we are unable to label several barriers. We attribute these missed detections to the ambiguity of the class name `barrier`. Notably, a `barrier` in nuScenes may not be the same as `barrier` as defined in internet pre-training data [18]. In (**d**), (**e**), and (**f**), we produce duplicate boxes for the same instances, indicating a failure of NMS.

**Significant Improvements in Low-Data Setting.** Pre-training CenterPoint with CM3D pseudo-labels and fine-tuning on labeled data improves over the prior art of PRC [9] by 8.1 mAP and 2.8 NDS with 5% labeled data. Similarly, pre-training BEVFusion with CM3D pseudo-labels outperforms prior art (CALICO) by 8.6 mAP and 4.6 NDS. Notably all pre-training strategies improve over random initialization (Rand. Init.). This suggests that aligning your pre-training and fine-tuning tasks can yield better performance. See Appendix C for additional results with 20% and 50% ground truth labels. Notably, we find that the impact of our proposed pre-training approach diminishes when training with additional ground truth data.

**Qualitative Results.** We visualize the output of our pseudo-label generator in Fig. 4. Although our method generates reasonable predictions in many cases, we find that our method fails in cases of occlusions (where there is no 3D information) and in cases where the VLM predicts a false positive detection with high confidence. See Fig 4 for detailed analysis and Appendix A and additional visuals.

## 5 Conclusion

In this paper, we propose Cross-Modal 3D Detection Distillation (CM3D), a zero-shot approach for generating 3D bounding boxes using vision-language models. We demonstrate that pre-training detectors with zero-shot 3D bounding box pseudo-labels achieves higher downstream semi-supervised detection than prior contrastive learning objectives. We conduct comprehensive experiments to ablate our design choices and demonstrate that our simple method significantly improves over prior work in limited data settings. However, we find that our proposed pre-training strategy limits generalization to arbitrary downstream tasks. Moreover, our current pseudo-labeling pipeline requires LiDAR for precise box localization and HD maps for orientation estimation, which may not be readily available for broader robotics domains (Appendix J).

## 6  Acknowledgements

We would like to thank Jenny Seidenschwarz for her feedback on early drafts. This work was supported in part by the NSF GRFP (Grant No. DGE2140739).

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

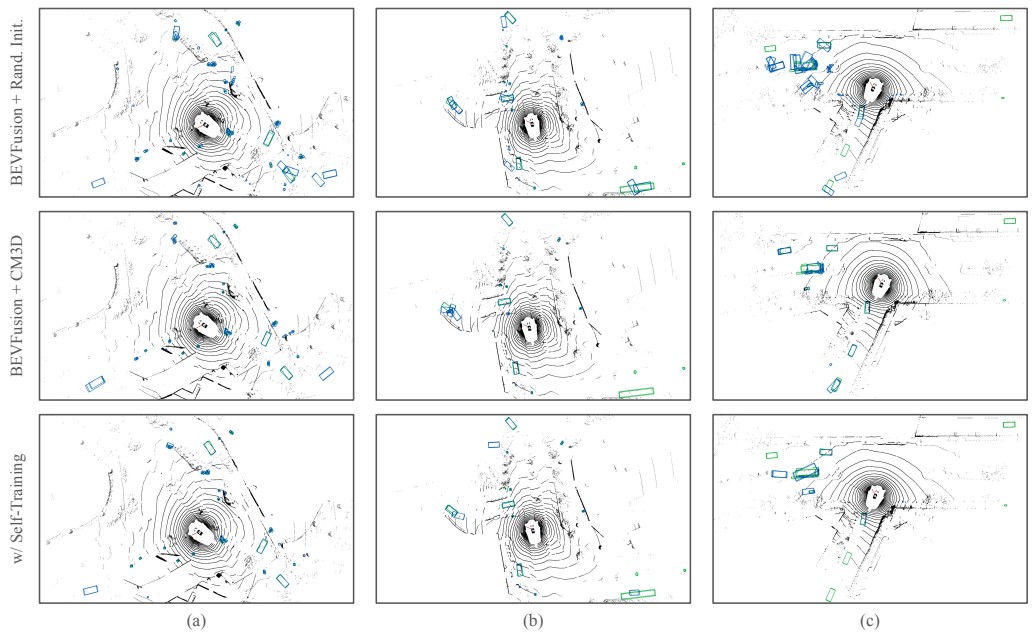

Figure 5: **More Qualitative Results**. We present additional qualitative results comparing the predictions from BEVFusion trained from scratch on 5% data (top), BEVFusion + CM3D (middle), and BEVFusion + CM3D w/ Self-Training (bottom). Ground truth bounding boxes are shown in green, and predictions are shown in blue. Across all three examples, we find that the model trained from scratch produces many high confidence false positives. Pre-training BEVFusion with CM3D pseudo-labels improves performance by reducing the number of false positives. However, many of the predictions have incorrect orientation estimates. Lastly, we find that self-training improves orientation estimation.

## A   More Qualitative Results

We present additional qualitative results comparing the predictions from BEVFusion trained from scratch on 5% data (top), BEVFusion + CM3D (middle), and BEVFusion + CM3D w/ Self-Training (bottom). Please see Figure 5 for detailed analysis.

## B   CM3D Pseudo-Label Refinement

Many components in our CM3D pipeline rely on data-driven priors and can only provide rough 3D estimates. We describe several strategies for improving our 3D psuedo-labels below.

**Prompt Engineering.** Although VLMs show impressive zero-shot performance, they struggle when the prompted class is different from concepts encountered in their training data [18]. Following prior work [11], we prompt Detic with the standard nuScenes class names and their synonyms (e.g. {human, adult, person, pedestrian} for class pedestrian, and {car, sedan, SUV} for class car). Specifically, we use the nuScenes annotator guide to understand how nuScenes defines each class and generate synonyms accordingly. As shown in Fig. 3, Detic predicts class names and 2D bounding boxes for each image, along with confidence scores for each detection. We then perform non-maximum suppression (NMS) to remove redundant predictions across synonyms. Interestingly, Detic is unable to accurately detect classes like barrier even with carefully designed prompts, suggesting that prompting with synonyms is insufficient for certain ambiguously defined classes [18].

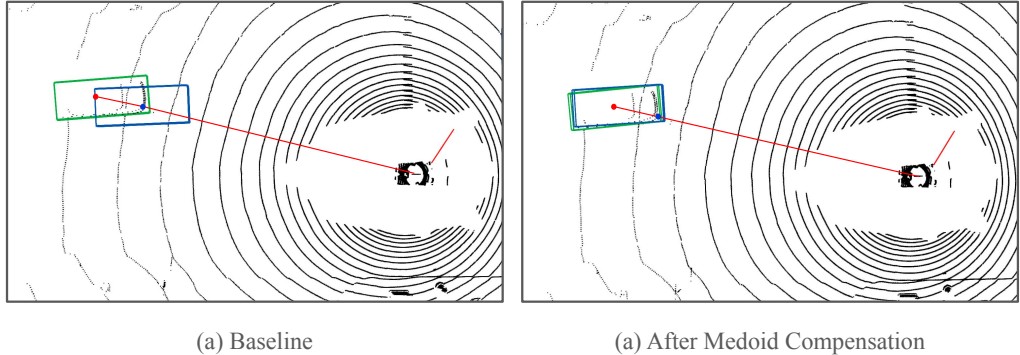

| (a) Baseline | (a) After Medoid Compensation |

Figure 6: **Medoid Compensation.** We find that all predicted medoids (on the **left**) tend to be radially biased toward the ego vehicle. This is because the LiDAR pointcloud only captures the visible surface of the car and not its full shape. To compensate for this bias, we "push" all predictions radially away (on the **right**), i.e., along the line connecting the center of the ego vehicle and the object medoid by a distance proportional to the object's size. Empirically, we show that this geometric trick improves mAP by 1.6% and NDS by 2.1%, respectively.

**Mask Erosion.** Although instance segmentations from SAM [56] are often accurate, we find that background LiDAR points near object boundaries can significantly impact medoid estimation [69]. We employ mask erosion to remove noisy LiDAR points near mask boundaries. These points are often unreliable because of depth discontinuities and minor errors in sensor calibration.

**LiDAR Accumulation.** LiDAR sweeps are notoriously sparse at range, making it difficult to distinguish foreground-vs-background. Therefore, the community has adopted the practice of accumulating multiple ego-motion compensated LiDAR sweeps when training 3D detectors [1]. We adopt the same practice in our pseudo-label generation pipeline for two reasons. First, accumulating multiple sweeps makes our medoid estimate more robust to outliers. Second, it biases the medoid towards the surface of the object, making medoid compensation (discussed next) more reliable.

**Medoid Compensation.** We find that predicted medoids are radially biased toward the ego vehicle because LiDAR points are denser on visible surfaces of objects as perceived from the ego vehicle. To compensate for this bias, we "push" all predictions radially away from the ego vehicle by a distance proportional to the object's size as follows:

Let $\vec{C}$ be the medoid of the object in the global coordinate frame, $\vec{E}$ be the center of the ego vehicle with respect to the global coordinate frame, and $\theta$ be the heading of the object in the global coordinate frame. We define a vector $\vec{CE} = \vec{E} - \vec{C}$, and $\alpha$ as the global slope angle of this vector, i.e., $\alpha = \arctan\left(\frac{\vec{CE}_y}{\vec{CE}_x}\right)$. As shown in Figure 6, we "push" the medoid back by distance $d = \min\left(\left|\frac{w}{2\sin(\alpha-\theta)}\right|, \left|\frac{l}{2\cos(\alpha-\theta)}\right|\right)$. Therefore, our new medoid is $\vec{C}'_x = \vec{C}_x - d \cdot \cos(\alpha)$ and $\vec{C}'_y = \vec{C}_y - d \cdot \sin(\alpha)$. We find that this simple geometric trick works surprisingly well in practice.

**Non-Maximum Suppression.** nuScenes uses six RGB cameras to capture a 360° view of the environment, where neighboring cameras capture overlapping regions. Naively generating pseudo-labels across cameras can produce repeated detections for the same instance. Therefore, we perform non-maximum suppression (NMS) in the overlapping regions [70] after medoid compensation to remove duplicate detections.

**Late Fusion.** Recall, we define the *center* of each predicted cuboid to be the medoid of the LiDAR points within an instance mask, the *dimensions* (length, width, height) as reported by ChatGPT when prompted with the class name, and the *orientation* to be aligned with lane geometry provided

by an HD map. Therefore, the quality of our pseudo-label generation pipeline is entirely dependent on the accuracy of our shape and orientation priors. In contrast, SAM3D [60] does not use priors for shape and orientation estimation, but rather directly estimates a rotated cuboid from a BEV perspective point cloud. Although SAM3D does not predict semantics, we find that its rotation and shape estimates are often more accurate than our priors. Therefore, we propose a simple late-fusion strategy to combine the best attributes of both zero-shot predictions.

For a given timestep, we greedily match our zero-shot predictions with SAM3D's predictions using 2D BEV IoU. Spatially matching CM3D and SAM3D predictions yields three categories of detections: matched detections, unmatched CM3D detections (without corresponding SAM3D detections), and unmatched SAM3D detections (without corresponding CM3D detections). We discard unmatched SAM3D detections since these are likely false positives because distinguishing foreground-vs-background with LiDAR-only cues is difficult [71].

Fusion of matched predictions from two independent detectors requires their scores to be comparable, Therefore, we use a class-agnostic implementation of score calibration as defined in [72]. Specifically, we scale the logits for SAM3D using a scaling factor $\tau$ (obtained by grid search on a val-set), i.e., confidence value $c = \sigma(logit/\tau)$. We construct a new set of fused detections by selecting the size and orientation from the more confident detection (SAM3D vs. CM3D) after calibration and use the semantic class predicted by CM3D (since CM3D can more accurately predict semantics with RGB images). Finally, we add all unmatched CM3D predictions to the set of fused predictions, unchanged.

**Implementation Details.** When generating pseudo-labels with CM3D, we use all Detic predictions with a confidence greater than 10% and use an IoU threshold of 0.75 for 2D NMS. We use a $3 \times 3$ kernel for mask erosion and accumulate the past 3 LiDAR frames for densification. Note that this is different from the usual 10-frame accumulation in nuScenes since 10-frame LiDAR pointcloud accumulation creates long "tails" for moving objects, and leads to inaccurate medoid predictions. For 3D NMS, we use class-specific distance-based thresholds defined in [62].

When training all detectors with pseudo-labels, we employ standard augmentation techniques (except for copy-paste augmentation, see Appendix E). Following established practices, we aggregate the past 10 sweeps for LiDAR densification using the provided ego-vehicle poses. We train CenterPoint and BEVFusion using the same hyperparameters prescribed by their respective papers. For CenterPoint, we first train the detector for 20 epochs with CM3D pseudo-labels and fine-tune the detector for 20 epochs using the limited set of annotations. For BEVFusion, we first pre-train the LiDAR-only branch using CM3D pseudo-labels for 20 epochs and fine-tune the LiDAR-only branch for 20 epochs using the limited set of annotations. We train the fusion model (RGB + LiDAR) for 6 epochs using the limited amount of labeled training data. Lastly, we fine-tune all models using self-training. Specifically, we use the fine-tuned model to generate new pseudo-labels on the unlabeled portion of the train-set and retrain the detector on the entire train-set (including the limited set of ground truth labels and pseudo-labels) from scratch. We ablate self-training further in Appendix D. We conduct all experiments on 8 RTX 3090 GPUs.

## C   Diminishing Returns with More Labeled Data

Although BEVFusion + CM3D shows significant performance improvements over CALICO in the low-data regime (e.g. 5% training data), the delta between both methods decreases as we increase the amount of training data (+8.6 mAP / +4.6 NDS at 5%, +3.3 mAP / +2.6 NDS at 10%, +1.4 mAP / +0.7 NDS at 20%, -0.3 mAP / -0.2 NDS at 50%). This suggests that the specific pre-training strategy matters less with increasing ground truth data.

## D   Ablation on Self-Training

To further understand the impact of self-training, we investigate two questions: (1) Does self-training provide any improvement over long-training schedules and (2) how does self-training improve NDS?

Table 4: **Semi-Supervised 3D Detection on nuScenes**. We evaluate the impact of CM3D pre-training with increasing annotation budget. We find that performance differences between methods (including random initialization) shrinks considerably. This suggests that the specific pre-training strategy matters less with increasing annotation budget. We highlight the best LiDAR-only (L) results in blue, and the best Camera-LiDAR (L+C) results in red.

| Training Data | Method | Modality | | mAP ↑ | NDS ↑ |
|---|---|---|---|---|---|
| | | Train | Test | | |
| | CenterPoint [62] (Rand. Init.) | L | L | 47.1 | 56.7 |
| | PointContrast [6] | L | L | 48.3 | 57.5 |
| | ProposalContrast [7] | L | L | 48.0 | 57.4 |
| | PRC [9] | L | L | 49.5 | 58.9 |
| 20% | CenterPoint [62] + CM3D (Ours) | L + C | L | 54.5 | 59.0 |
| | BEVFusion [63] (Rand. Init.) | L + C | L + C | 53.1 | 58.9 |
| | SimIPU [8] | L + C | L + C | 53.4 | 58.9 |
| | PRC [9] + BEVDistill [68] | L + C | L + C | 54.4 | 59.2 |
| | CALICO [9] | L + C | L + C | 54.8 | 59.5 |
| | BEVFusion [63] + CM3D (Ours) | L + C | L + C | 56.2 | 60.2 |
| | CenterPoint [62] (Rand. Init.) | L | L | 53.2 | 61.0 |
| | PointContrast [6] | L | L | 53.5 | 61.4 |
| | ProposalContrast [7] | L | L | 53.1 | 61.0 |
| | PRC [9] | L | L | 54.1 | 62.1 |
| 50% | CenterPoint [62] + CM3D (Ours) | L + C | L | 56.9 | 61.0 |
| | BEVFusion [63] (Rand. Init.) | L + C | L + C | 58.5 | 61.8 |
| | SimIPU [8] | L + C | L + C | 58.6 | 62.0 |
| | PRC [9] + BEVDistill [68] | L + C | L + C | 59.6 | 62.3 |
| | CALICO [9] | L + C | L + C | 60.1 | 62.7 |
| | BEVFusion [63] + CM3D (Ours) | L + C | L + C | 59.7 | 62.5 |

**Self-Training Algorithm**. As shown in Fig 7, given K% annotated training data and (1-K%) unlabeled training data with CM3D pseudo-labels, we first pre-train a randomly initialized detector on (1-K%) pseudo-labels, fine-tune on K% annotated data, and use the resulting fine-tuned model to re-label the (1-K%) unlabeled training data. We iterative refine the (1-K%) pseudo-labels through multiple rounds of self-training. Importantly, we randomly initialize the detector after each round of self-training and pseudo-label generation to simplify training. We find that even one round of self-

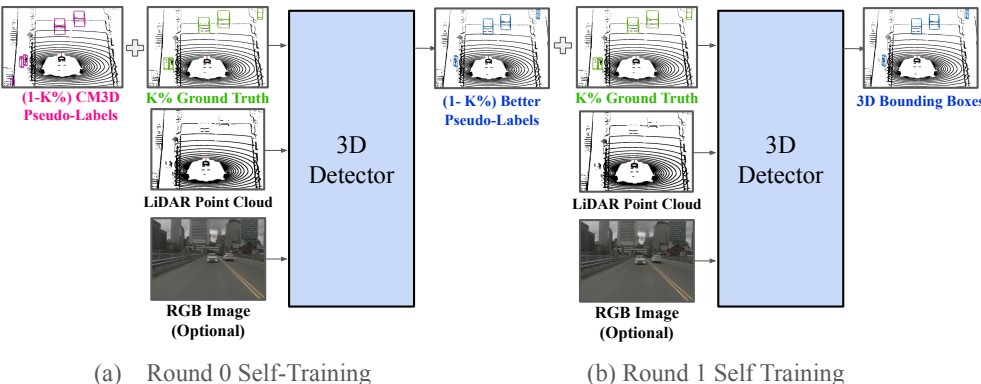

(a)    Round 0 Self-Training          (b) Round 1 Self Training

Figure 7: **Self-Training Procedure**. We train a detector with ground truth annotations from K% of the train-set an pseudo-labels from (1-K%) of the train-set. We iteratively use the trained detector to predict better pseudo-labels on the (1-K%) subset until the accuracy of the predicted detections plateaus. We visualize one round of self-training above.

Table 5: **Self-Training vs. Longer Training Schedule**. We evaluate the impact of training BEV-Fusion + CM3D with a longer-training schedule (2x Schedule, 3x Schedule, and 4x Schedule) and with multiple rounds of self-training (R1, R2, and R3). We note that one round of self training (R1) improves over BEVFusion + CM3D w/ 4x Schedule, suggesting that self-training provides greater benefit than simply training for longer. Further, additional rounds of self-training (R2 and R3) provides modest, but diminishing improvements.

| Training Data | Method | mAP ↑ | NDS ↑ |
|---|---|---|---|
| | SimIPU [8] | 39.1 | 45.8 |
| | PRC [9] + BEVDistill [68] | 41.0 | 47.5 |
| | CALICO [9] | 41.7 | 47.9 |
| | BEVFusion [63] + CM3D w/ 1x Schedule | 48.6 | 47.8 |
| 5% | BEVFusion [63] + CM3D w/ 2x Schedule | 49.3 | 49.5 |
| | BEVFusion [63] + CM3D w/ 3x Schedule | 49.9 | 50.4 |
| | BEVFusion [63] + CM3D w/ 4x Schedule | 50.3 | 51.1 |
| | BEVFusion [63] + CM3D w/ 1x Schedule + R1 Self-Training | 50.8 | 52.2 |
| | BEVFusion [63] + CM3D w/ 1x Schedule + R2 Self-Training | 51.1 | 52.5 |
| | BEVFusion [63] + CM3D w/ 1x Schedule + R3 Self-Training | 51.3 | 52.6 |

Table 6: **Analysis NDS Sub-Metric Errors**. We find that all sub-component metrics of NDS are improved with three rounds of self-training. Notably, improved mAP, and reduced orientation, attribute, and velocity estimation errors contribute the most to the NDS results. Surprisingly, self-training does not significantly improve size estimation, suggesting that our ChatGPT shape priors are relatively robust.

| Metric (%) | BEVFusion [63] + CM3D w/o Self-Training | w/ R3 Self-Training |
|---|---|---|
| mAP ↑ | 48.6 | **51.3** |
| mATE ↓ | 33.8 | **32.2** |
| mASE ↓ | 26.8 | **26.2** |
| mAOE ↓ | 60.7 | **41.1** |
| mAVE ↓ | 136.8 | **100.3** |
| mAAE ↓ | 43.0 | **30.2** |
| NDS ↑ | 47.8 | **52.6** |

training significantly improves NDS when fine-tuning detectors with limited annotations. Additional rounds of self-training provide limited improvements.

**Self-Training vs. Longer Training Schedules**. We compare the performance of BEVFusion w/ CM3D trained with a 1x schedule (20 epochs LiDAR-only branch pre-training with CM3D pseudo-labels, 20 epochs LiDAR-only branch fine-tuning with K% annotated data, and 6 epochs multi-modal fine-tuning with K% annotated data), 2x schedule, 3x schedule, and 4x schedule in Table 5. Although performance improves when training detectors for longer, self-training (even for one round) significantly improves more than longer training schedules.

**Self-Training Improves All Components of NDS**. Next, we compare the NDS sub-metrics for BEVFusion + CM3D with and without self-training in Table 6. Notably, we find that self-training improves *all* sub-metrics. We posit that self training improves classification accuracy and contributes to better mAP and lower attribute error. Similarly, we posit that self-training reduces pseudo-label bias for orientation and velocity estimation.

# E   Ablation on Copy-Paste Augmentation

State-of-the-art 3D detectors are often trained with copy-paste augmentation to improve detection accuracy for rare classes like `bicycle` or `construction vehicle`. Specifically, rare instances are pasted onto a LiDAR sweep to artificially increase the number of objects. Since our cross-modal 3D detection distillation pipeline creates an explicit 3D bounding box, we can easily apply copy-paste augmentation during pre-training as well. However, given that our pseudo-labels are

Table 7: **Ablation on Copy-Paste Augmentation.** We evaluate different permutations of training CenterPoint with and without copy-paste augmentation during pre-training and fine-tuning. Importantly, we find that turning off copy-paste augmentation during pre-training and turning it on during fine-tuning achieves the best performance.

| Pre-train
w/ Copy-Paste Augmentation | Fine-tune
w/ Copy-Paste Augmentation | Modality | mAP ↑ | NDS ↑ |
|:---:|:---:|:---:|:---:|:---:|
| ✗ | ✗ | L | 45.3 | 45.6 |
| ✗ | ✓ | L | 46.7 | 46.1 |
| ✓ | ✗ | L | 46.1 | 46.4 |
| ✓ | ✓ | L | 44.7 | 44.5 |

noisy, is copy-paste augmentation worth it? We train CenterPoint with and without copy-paste augmentation during pre-training and fine-tuning (on 5% of the training data) to ablate its impact. We find that turning augmentation off during pre-training and turning augmentation on during fine-tuning yields the best mAP. Notably, using copy-paste augmentation for both pre-training and fine-tuning performs the worst. Intuitively, copy-pasting noisy pseudo-labels will decrease the signal-to-noise ratio, leading to worse initialization. In contrast, copy-paste augmentation during fine-tuning improves performance in limited-data regimes because we are effectively increasing the size of the dataset.

# F   Replacing Detic with GroundingDINO

We ablate the impact of different VLMs on pseudo-label quality and downstream detection accuracy. Recall, our pseudo-label generation pipeline first prompts a VLM detector with class names (e.g. `car, bus, truck`) to generate 2D box proposals. We switch out Detic [15] for GroundingDINO [61] in Table 8. First, we find that CM3D w/ Detic generates better pseudo-labels than CM3D w/ GroundingDINO (23.0 mAP vs. 18.0 mAP). Surprisingly, we find that models pre-trained with Detic-based pseudo-labels achieve higher mAP, while models pre-trained with GroundingDINO-based pseudo-labels achieve higher NDS.

Table 8: **Comparison Between VLMs for Pseudo-Label Generation.** CM3D w/ Detic generates better pseudo-labels than CM3D with GroundingDINO (23.0 mAP vs. 18.0 mAP). Surprisingly, we find that models pre-trained with Detic-based pseudo-labels achieve higher mAP, while models pre-trained with GroundingDINO-based pseudo-labels achieve higher NDS.

| Training Data | Method | Modality | mAP ↑ | NDS ↑ |
|:---:|:---|:---:|:---:|:---:|
| 0% | SAM3D [60] | L | 1.7 | 2.4 |
| | CM3D w/ Detic (Ours) | L + C | 23.0 | 22.1 |
| | CM3D w/ GroundingDINO (Ours) | L + C | 18.0 | 19.7 |
| | CenterPoint [62] + CM3D w/ Detic (Ours) | L | 16.7 | 21.4 |
| | CenterPoint [62] + CM3D w/ GroundingDINO (Ours) | L | 16.0 | 21.2 |
| 5% | CenterPoint [62] + Rand. Init. | L | 33.1 | 37.4 |
| | CenterPoint [62] + CM3D w/ Detic (Ours) | L | 46.1 | 46.4 |
| | CenterPoint [62] + CM3D w/ GroundingDINO (Ours) | L | 42.9 | 52.3 |
| 10% | CenterPoint [62] + Rand. Init. | L | 41.1 | 48.0 |
| | CenterPoint [62] + CM3D w/ Detic (Ours) | L | 50.8 | 49.2 |
| | CenterPoint [62] + CM3D w/ GroundingDINO (Ours) | L | 49.8 | 56.7 |

# G   Ablation on Scaling Up Pseudo-Labels

We demonstrate how our method scales with an increasing amount of unlabeled data (shown as a percentage of the full nuScenes train-set) with a fixed budget of 5% labeled data in Fig. 8. For simplicity, we do not perform self-training.

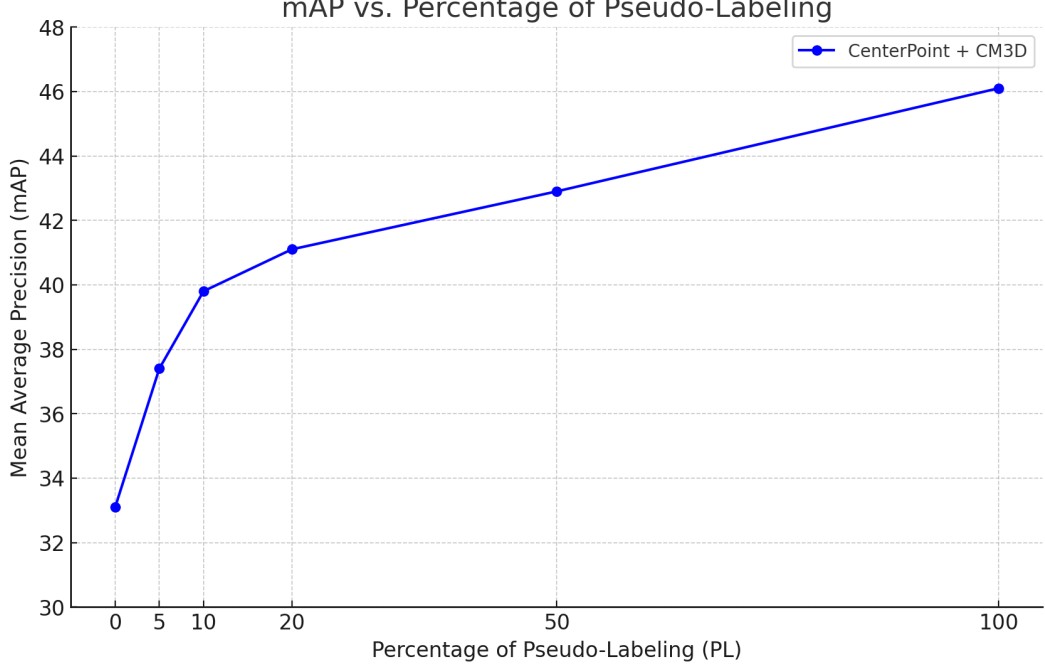

Figure 8: **Scaling Up Pseudo-Labels**. We plot the impact of scaling up the amount of unlabeled data on the nuScenes dataset on semi-supervised detection performance (mAP). Increasing the number of pseudo-labels (PL, shown as a percentage of the nuScenes train-set) used for pre-training significantly improves semi-supervised detection accuracy. This suggests that our method will continue to improve with larger sets of unlabeled datasets.

## H  Comparing ChatGPT Priors vs. Real Shape Priors

We determine the shape (e.g. width, length, and height) of a class by prompting a large language model (ChatGPT) with the following:

"What are the average sizes in meters (values can be floats) of the following object classes? Give the answer in the form of a JSON file using the following format: [width (side to side), length (front to back), height]
Object classes: car, truck, bus, trailer, construction_vehicle, pedestrian, motorcycle, bicycle, traffic_cone, traffic_barrier.
Do not answer with anything other than the JSON output."

Despite not having access to our specific 3D training data, LLMs have seen many descriptions of object shapes in their training data and can offer plausible 3D sizes for common objects. We compare shape priors from ChatGPT with anchor boxes derived from the training data in Table 9.

## I  Evaluating CM3D on Waymo

We evaluate our method on the Waymo Open Dataset [2] in Table 10. Following prior work, we report mAP [65] and mAPH [2] (a custom metric proposed by Waymo that incorporates heading). For fair comparison with SAM3D [60], we only evaluate predictions between 0 and 30 meters from the ego-vehicle. Notably, SAM3D achieves significantly better performance on Waymo than nuScenes, likely due to Waymo's greater point density. Importantly, SAM3D only produces class-agnostic boxes and assigns the class `Vehicle` to all predictions. Therefore, it achieves 0 AP for both `Pedestrian` and `Cyclist`. In contrast, CM3D explicitly predicts semantic classes using RGB

Table 9: **Chat-GPT Shape Prior**. We compare the average 3D shapes of objects in nuScenes (denoted by {Width, Length, Height}) with predicted 3D shapes from ChatGPT. Impressively, Chat-GPT provides reasonable 3D object extents.

| Class Name | Real Shape Prior | ChatGPT Shape Prior |
|---|---|---|
| Car | {1.91, 4.62, 1.68} | {1.80, 4.50, 1.50} |
| Truck | {2.38, 6.89, 2.60} | {2.60, 8.00, 3.60} |
| Bus | {2.59, 11.47, 3.81} | {2.50, 12.00, 4.00} |
| Trailer | {2.29, 10.20, 3.70} | {2.60, 12.00, 3.60} |
| Construction Vehicle | {2.47, 5.5, 2.38} | {2.00, 4.50, 2.50} |
| Pedestrian | {0.60, 0.73, 1.76} | {0.40, 0.70, 1.70} |
| Motorcycle | {0.76, 1.95, 1.57} | {0.80, 2.10, 1.70} |
| Bicycle | {0.63, 1.82, 1.39} | {0.60, 1.80, 1.40} |
| Traffic Cone | {0.42, 0.43, 0.70} | {0.30, 0.30, 0.70} |
| Barrier | {0.60, 2.32, 1.06} | {0.50, 1.20, 0.90} |

images. However, we find that our approach only achieves marginal improvement over SAM3D. We attribute this to Waymo's evaluation metric and sensor setup.

Unlike nuScenes, which uses a distance-based threshold to match predictions with ground truth boxes when computing mAP, Waymo uses a stricter matching criteria based on 3D IoU with high thresholds of 0.7 for `Vehicle` and 0.5 for other classes. Recall that our size estimates are from ChatGPT and our orientation estimates are derived from an HD map. These imprecise estimates significantly harm detection accuracy since high IoU between predictions and ground truth boxes requires precise size and orientation. Furthermore, the Waymo Open Dataset does not include rear-facing cameras, making it impossible for CM3D to predict bounding boxes behind the ego-vehicle. As a result, we find that our method is heavily dependent on late-fusion with SAM3D, which is able to correct the size and orientation of CM3D pseudo-labels, and generate predictions for parts of the scene without RGB information.

We pre-train CenterPoint with our pseudo-labels for 30 epochs and fine-tune the model for 30 epochs. We find that pre-training with CM3D pseudo-labels consistently improves over random initialization. However, we find that our method performs slightly worse than prior LiDAR-only methods like PRC. We posit that the lack of RGB camera coverage for more than 50% of each LiDAR sweep, and our reliance on SAM3D's class-agnostic pseudo-labels significantly diminishes the benefit of our proposed approach.

Table 10: **Zero-Shot Waymo 3D Detection.** CM3D marginally improves over SAM3D's LiDAR-only predictions even though it has access to both RGB and LiDAR. However, pre-training a LiDAR-only detector (CenterPoint) on CM3D psuedo-labels extracted from the train-set significantly improves accuracy, illustrating the benefit of cross-modal learning. Importantly, we note that lower performance for `pedestrian` and `cyclist` can be attributed to the stricter matching criteria used by WOD. Specifically, a detection is considered a true positive only if it overlaps with the ground truth with 3D IOU greater than 0.7 for `vehicle` and 0.5 for other classes.

| Method | Test Modality | Vehicle | | Pedestrian | | Cyclist | |
|---|---|---|---|---|---|---|---|
| | | AP ↑ | APH ↑ | AP ↑ | APH ↑ | AP ↑ | APH ↑ |
| SAM3D | L | 19.1 | 13.0 | 0.0 | 0.0 | 0.0 | 0.0 |
| CM3D (Ours) | L + C | 19.4 | 13.4 | 0.2 | 0.1 | 0.7 | 0.5 |
| CenterPoint + CM3D (Ours) | L | 23.7 | 17.5 | 0.1 | 0.1 | 2.6 | 1.3 |

Table 11: **Waymo 3D Detection Results**. We report Level-2 mAP and mAPH averaged over 3 classes (`Vehicle`, `Pedestrian`, and `Cyclist`). CM3D consistently improves over random initialization. However, we find that our method performs slightly worse than prior LiDAR-only methods like PRC.

| Training Data | Method | Modality | | mAP ↑ | mAPH ↑ |
|---|---|---|---|---|---|
| | | Train | Test | | |
| | SAM3D | L | L | 6.4 | 4.3 |
| 0% | CM3D (Ours) | L + C | L + C | 6.8 | 4.7 |
| | CenterPoint [62] + CM3D (Ours) | L + C | L | 8.8 | 6.2 |
| | SECOND [73] + Rand. Init. | L | L | 53.1 | 49.1 |
| | FixMatch [74] (SECOND Backbone) | L | L | 55.8 | 51.5 |
| | ProficientTeachers [75] (SECOND Backbone) | L | L | 58.6 | 54.2 |
| | CenterPoint [62] + Rand. Init. | L | L | 63.2 | 61.0 |
| 20% | PointContrast [6] | L | L | 65.2 | 62.6 |
| | ProposalContrast [7] | L | L | 66.3 | 63.7 |
| | PRC [9] | L | L | 68.6 | 65.5 |
| | CenterPoint [62] + CM3D (Ours) | L + C | L | 67.6 | 64.4 |

# J   Limitations and Future Work

We propose a simple cross-modal detection distillation framework that leverages paired multi-modal data and vision-language models to generate zero-shot 3D bounding boxes. We demonstrate that pre-training 3D detectors with our zero-shot 3D bounding boxes yields strong semi-supervised detection accuracy. We discuss several limitations of our approach below.

**Limitation: Aligning Pre-Training and Fine-Tuning Task Limits Generalizability**. Contrastive learning has been widely adopted for self-supervised learning because it encodes task-agnostic representations that can be used for diverse downstream applications. In contrast, our approach uses prior knowledge about the downstream task to design a suitable pretext task. While this works well when the pre-training and fine-tuning tasks are well aligned, it does not provide a significant improvement when this is not the case. We evaluate the generalization of BEVFusion pre-trained on CM3D pseudo-labels for BEV map segmentation. Surprisingly, our pre-training strategy performs better at BEV map segmentation than random initialization! However, our method performs slightly worse than other state-of-the-art self-supervised methods methods. This suggests that aligning our pre-training and fine-tuning task can provide significant improvements (cf. Table 3 in the main paper) at the cost of generalizability to diverse tasks. Future work should explore different shelf-supervised pretext tasks to improve semi-supervised accuracy for diverse tasks in low data settings.

**Limitation: Orientation Estimation Requires HD Maps**. Our proposed approach uses lane direction from HD maps to estimate vehicle orientation. This heuristic does not work well when vehicles are turning into an intersection, for non-vehicles, and when HD maps are unavailable. Instead, future work should explore using multi-object trackers to generate heading estimates from consecutive detections [39]. We posit that this can improve NDS, and can potentially eliminate the need for self-training.

**Limitation: Data Sampling Strategy**. Although our semi-supervised experiments follow the suggested protocol in [7, 49, 9] and sample K% of the training data uniformly from the entire training set, this may be unrealistic in practice. Sampling training data uniformly artificially inflates the diversity of training samples and is more time consuming to annotate than sampling training data from consecutive frames. Future work should explore the semi-supervised setting with data sampled from consecutive frames.

**Future Work: Combining Pretext Tasks to Improve Performance.** Since our pre-training approach is entirely disjoint contrastive pre-training methods, we hypothesize that our pre-training setup can be used in tandem with such methods to make more accurate predictions and improve results. This may provide a middle-ground between task-agnostic contrastive learning and task-specific pseudo-label pre-training.

Table 12: **nuScenes BEV Map Segmentation Results**. Although BEVFusion + CM3D is pre-trained on noisy 3D bounding boxes, it performs better on BEV map segmentation than random initialization! However, our method performs worse than other state-of-the-art methods. This suggests that aligning pre-training and fine-tuning tasks improves semi-supervised performance for the target task (cf. Table 3) at the cost of generalizing to other downstream tasks.

| Training Data | Method | mIOU ↑ |
|---|---|---|
| 5% | BEVFusion [63] + Rand. Init. | 36.3 |
| | SimIPU [8] | 38.5 |
| | PRC [9] + BEVDistill [68] | 40.9 |
| | CALICO [9] | **42.0** |
| | BEVFusion [63] + CM3D (Ours) | 38.5 |
| 10% | BEVFusion [63] + Rand. Init. | 43.8 |
| | SimIPU [8] | 45.1 |
| | PRC [9] + BEVDistill [68] | 46.4 |
| | CALICO [9] | **47.3** |
| | BEVFusion [63] + CM3D (Ours) | 44.4 |

**Future Work: Application to Broader Robotics Tasks.** Our approach leverages domain-specific insights, and may not be directly generalizable to robotics applications without LiDAR or HD Maps. Future work can adapt our pseudo-labeling framework for such situations. For example, instead of using LiDAR, one can directly use metric depth estimates [76] from state-of-the-art monocular depth estimators. Similarly, one can estimate orientation with rotating calipers or track-level information. We present preliminary results of our pseudo-labeler applied to non-AV applications in Fig. 9.

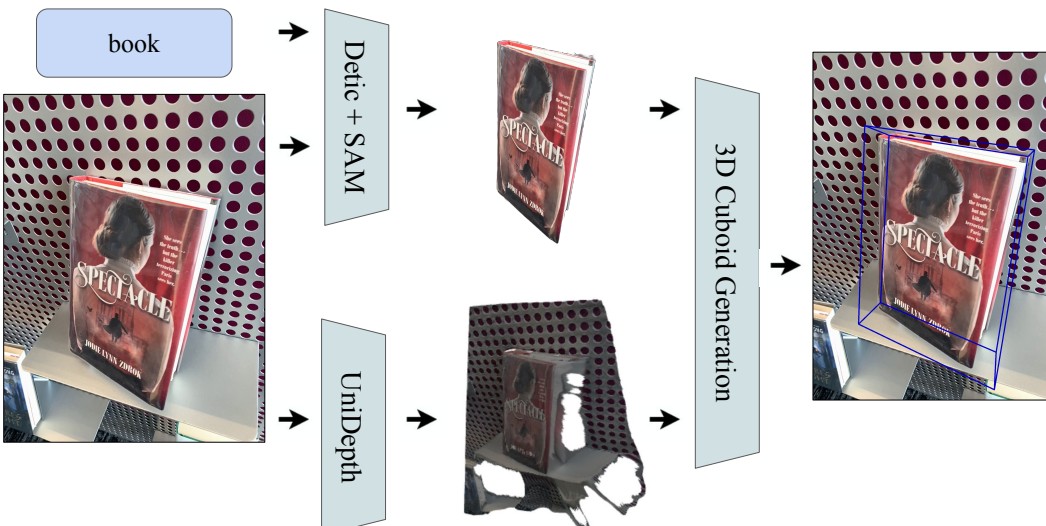

Figure 9: **Generalized Pseudo-Labeler**. We visualize the output of our generalized psuedo-labeler on several images above. Importantly, we don't assume we have access to LiDAR for precise depth estimates or HD maps for orientation estimation. Instead, we use UniDepth [76] to generate a pseudo-point cloud and estimate orientation using rotating calipers. Importantly, orientation estimates can be up to 180 degrees off because this current setup cannot estimate heading. This suggests that our pseudo-labeler can generalize beyond autonomous vehicle settings with minor modifications. We expect that semi-supervised training can further improve the quality of predicted bounding boxes.

