# OpenReview forum: "Shelf-Supervised Cross-Modal Pre-Training for 3D Object Detection"
_robot-learning.org/CoRL/2024/Conference — CoRL 2024_

### Official Review · Reviewer_yS2F · 2024-07-18
**This paper introduces an innovative approach to 3D object detection using vision-language models for pre-training with pseudo-labels, promising for reducing annotation needs while raising concerns about accessibility and generalization.**

**Originality:** 3
**Technical Quality:** 3
**Clarity Of Presentation:** 4
**Potential Impact:** 2
**Recommendation:** 2
**Confidence:** 4

**Review:**

Strengths
S1: The task of transferring knowledge from 2D foundation models to 3D detection is both valuable and effective.

S2: The authors propose a series of tricks to enhance the quality of pseudo-labels, which is a critical component of their method. These include prompt engineering, mask erosion, LiDAR accumulation, medoid compensation, and late fusion. These techniques demonstrate a deep understanding of the challenges in 3D detection and provide practical solutions to improve the accuracy of the generated pseudo-labels.

S3: The paper is well-written and includes extensive ablation studies and experiments to prove the effectiveness of their approach. The authors have conducted thorough experiments on the nuScenes and Waymo Open Dataset, showing significant improvements over prior work in limited data settings.

Weaknesses
W1: While the model can be deployed in autonomous vehicles, its relevance to the broader field of robotics is less clear. The focus on LiDAR and vision-based detection may limit its applicability to other types of robotic perception tasks that rely on different sensor modalities or operate in environments where such sensors are not available.

W2: A major concern is the performance of the proposed model in fully supervised settings. Due to the additional cross-modal training data, CM3D demonstrates superior performance in semi-supervised learning, particularly with 5% of the training data. However, it falls short of state-of-the-art (SOTA) models under 50% of the NuScenes data and 30% of the WOD data. This raises concerns about its efficacy in fully supervised learning, which may hinder the impact of the paper.

**Quality Of The Limitations Section:**

3

**Questions For Rebuttal:**

Q1: The authors only validate their approach on NuScenes and WOD datasets. Why not test on KITTI or even indoor dataset like ScanNet for a broader evaluation?

Q2: The distillation approach from 2D foundation models to 3D is commendable, especially leveraging unlabeled autonomous driving data. Why not explore training with more unlabeled data, such as a combination of multiple datasets?

**Robotics Focus:**

2

**Summary Of Paper:**

The paper presents a compelling approach to 3D object detection by leveraging vision-language models to generate pseudo-labels for pre-training. The method is innovative, well-evaluated, and shows promise for reducing the annotation burden in 3D detection tasks.  However, its reliance on multiple data modalities and external models could limit its accessibility and generalization capabilities. In addition, the author proposed a series of tricks to improve the accuracy of pseudo-labels and carried out a large number of very solid experiments. Overall, the work contributes valuable insights to the field of 3D object detection.

**Summary Of Recommendation:**

Despite the limited performance and innovation of this paper, I am inclined to accept it due to the thorough ablation experiments and analysis.

---

### Official Review · Reviewer_rkuQ · 2024-07-21
**Not in the scope of CoRL submission**

**Originality:** 3
**Technical Quality:** 3
**Clarity Of Presentation:** 3
**Potential Impact:** 2
**Recommendation:** 1
**Confidence:** 5

**Review:**

As stated in the review guideline, this paper is clearly not relevant to a physical or simulated robot. I suggest the authors submit to a more related conference.

Please check the following guideline from https://www.corl.org/contributions/instruction-for-reviews

Scope: All CoRL submissions must demonstrate the relevance to Robot Learning through

Intent: Explicitly address a learning question for physical robots OR

Outcome: Test the proposed learning solution on physical robots.

Rejection examples

No learning: Manually design and tune the performance of a robot controller without use of learning.

No learning: A search algorithm for model-based planning.

No robotics: A generic result on sample complexity.

No robotics: A generic RL algorithm.

Little robotics: Improved performance on a standard CV dataset, e.g., ImageNet recognition.

Gray

An RL algorithm that works only in simulator X. Does it transfer to real robot learning (sim2real, data efficiency, …)? Yes for CARLA for autonomous driving. No for Cheetah/Human-oid in Mujuco. According to our stated principles, the submission satisfies the intent. Its failure or success to demonstrate the relevance will be determined during the review process.

**Quality Of The Limitations Section:**

2

**Questions For Rebuttal:**

N/A

**Robotics Focus:**

1

**Summary Of Paper:**

The paper proposes a multi-modal pretraining method for 3D object detection, however, it is not related to the Robotics, I suggest it should be submitted to a CV conference.

**Summary Of Recommendation:**

Not relevant to robotics

---

### Official Review · Reviewer_XAhw · 2024-08-03
**lack of originality**

**Originality:** 1
**Technical Quality:** 3
**Clarity Of Presentation:** 3
**Potential Impact:** 2
**Recommendation:** 1
**Confidence:** 4

**Review:**

**Strengths**
- The paper is fairly presented and the main idea is easy to follow in the writing
- Good results improving the number on the previous SOTA method in this setting

**Weakness**

**[Major]**
 - I fail to understand where the main contribution of this paper lies.
Using a VLM + SAM to segment out the object masks should not be the main contribution as these are well-known tools. Combining them with the 3D data well should be crucial in getting good quality pseudo-labels but 2d-to-3D projection relies on hardcoded heuristics (mostly derived from [1]) which do not seem generalisable.

**[Minor]**
- The experiment section can be made clearer by mentioning the choice of baselines and explaining their significance to the readers. For eg: it's hard to understand what is the rand. Initialisation baseline for your weakly supervised experiments

**References**

[1] B. Wilson, Z. Kira, and J. Hays. 3d for free: Crossmodal transfer learning using hd maps

**Quality Of The Limitations Section:**

2

**Questions For Rebuttal:**

none (see weaknesses)

**Robotics Focus:**

2

**Summary Of Paper:**

This paper proposes an easy to use pipeline for labelling 3D datasets utilising off-the shelf foundation models . They generate bounding box proposals in the image space using a VLM and then pass it to SAM to get the segmented out objects. The object segments are converted to 3D box labels using pre-defined priors over the common objects of interest and 3D maps in self-driving. The authors evaluate on 2 self-driving datasets (NuScenes and Waymo) and show that pertaining on these pseudo labels boosts performance on the downstream detection tasks.

**Summary Of Recommendation:**

Combining foundational models in different modalities such as VLM, SAM, or SAM3D (for LiDAR data) to generate pseudo-labels is an interesting and useful problem. However, this work approaches it in a relatively straightforward manner, mostly relying on [1]. Despite the performance gains, the results also fail to provide any insight into this problem for the community at this point.

---

### Official Review · Reviewer_gB95 · 2024-08-04
**Good results, well motivated, some minor writing issues**

**Originality:** 3
**Technical Quality:** 4
**Clarity Of Presentation:** 3
**Potential Impact:** 3
**Recommendation:** 3
**Confidence:** 3

**Review:**

## Strengths

- Writing is clear and overall the paper and method is easy to understand
- The method is neat and seems clearly addresses the motivation defined in the intro: CM3D performs best on low-data regimes to help pre-train 3d object detectors.
- A lot of comprehensive comparisons/ablations in the appendix
- The authors demonstrate that CM3D works well as a label generator for two separate baselines

## Weaknesses

- The self-training part is glossed over, but there should be some high-level details in the main paper that highlight what the inputs/outputs of the self-training procedure are, and/or a more clear figure for it.
- There’s a lot of great additional experiments in the supplementary, but these are not detailed extensively in the main paper. Some relevant appendix section links from the main paper (where relevant) to each of the additional experiment sections would result in less digging through the appendix. This could be solved by having a paragraph on “ablation studies”/”additional experiments” in the experiments section.
- There are very few limitations talked about in the main paper, it’d be better to put the limitations section explicitly in the main.
- One thing that’s unclear is: How reliant is the method on the ChatGPT generated shape priors? What if some noise was introduced in the shape priors, would the performance drastically change?
- SAM is used to generate object masks from bounding boxes. This doesn’t seem to be ablated, but it is an extra step which requires a lot of compute that I’m not sure is necessary. Why can’t the 2D-to-3D projection block simply take the bounding box as input to generate the pseudo-labels?

**Quality Of The Limitations Section:**

2

**Questions For Rebuttal:**

See weaknesses for questions, along with some additional questions that are not related to paper weaknesses:

- 3D detection labels are also important in other subfields, like in robotic manipulation, where many works have recently demonstrated that 3D policies perform and generalize better than 2D policies. This approach does seem a little engineered for self-driving specifically; how applicable would this be to, e.g., robot manipulation?

**Robotics Focus:**

2

**Summary Of Paper:**

The paper aims to generate pseudo-labels for 3D driving data to generate 3D labels to train 3D object detection models. It does so by using ChatGPT-generated shape priors and pre-trained 2D object detection models to automatically construct 3D bounding boxes.

**Summary Of Recommendation:**

The paper is overall well written and has comprehensive experiments.

---

### Author Rebuttal · Authors · 2024-08-07

Reviewers find that our paper presents a “compelling approach to 3D object detection by leveraging vision-language models to generate pseudo-labels for pre-training”. Reviewer yS2F finds that our method is “innovative, well evaluated and shows promise”, and “contributes valuable insights to the field of 3D object detection”. Further, Reviewer XAhw highlights that we address an “interesting and valuable problem”. Lastly, Reviewer gB95 appreciates that the “paper and method is easy to understand”. We address major concerns below.

- Reviewer gB95 highlights that “there’s a lot of great additional experiments in the supplementary, but these are not detailed extensively in the main paper”. We will include forward pointers in the main paper to relevant experiments in the supplement.
- Reviewer XAhw “fails to understand the main contribution of this paper” and claims that our paper lacks originality. Our pseudo-labeling pipeline modernizes Wilson et. al. using recent advances in vision foundation models. Unlike Wilson et. al., we do not use any dataset specific models for frustum proposals (and instead use VLMs) or dataset-specific annotations for shape priors (and instead use LLMs). Importantly, the primary contribution of our work is in using our SOTA pseudo-labels (Appendix D) to improve the label-efficiency of 3D object detectors. Our key insight is that pre-training 3D detectors with high quality pseudo-labels outperforms prior contrastive learning approaches in limited annotation settings (Table 1).
- Reviewer rkuQ is concerned that our paper is not in scope for CoRL. We disagree because prior accepted papers at CoRL also address 3D detection for autonomous vehicles.
- Reviewer yS2F is concerned about our model performance in fully supervised settings. We agree that the improvement of our pre-training strategy diminishes with more labeled data (L232). However, we focus on the limited annotation setting since the impact of pre-training diminishes with sufficient labeled data in general.

Paper Updates
- gB95: We've added a figure describing the self-training procedure in Appendix C
- yS2F: We've added additional results highlighting how our method scales with more unlabeled data in Appendix D.
- gB95, XAhw, yS2F: We added a discussion on how to adapt our current setup for broader robotics applications in Appendix K. We also provide preliminary visuals of our pseudo-labeler applied to non-autonomous vehicle data.

We provide a revised draft below.

---

### Decision · Program_Chairs · 2024-09-05

**Decision:**

Accept

**Comment:**

The paper presents a method (CM3D) for training 3D object detection models by utilizing pseudo-labels for 3D driving data -- constructing 3D bounding boxes with ChatGPT-generated shape priors and pre-trained 2D detection models. The proposed technique is shown to be particularly effective in low-data regimes and shows promising results in enhancing the training of 3D object detectors. Extensive comparisons and ablations detailed in the appendix validate the approach, demonstrating its efficacy across two self-driving datasets, NuScenes and Waymo. At the same time, some reviewers pointed out that the novelty of the proposed technique is relatively weak given previous SOTA models, and some other weaknesses, such as insufficient details on the self-training process and the reliance on computationally intensive steps like SAM for object masking. Despite the authors response, the reviewers didnot change their scores. The method has strong potential is well-studied and the potential applicability to other areas (e.g., robotic manipulation). However, the paper warrants a re-review after authors refine and expand on the method's limitation in the main paper. We acknowledge that one of the reviewer claimed that the paper is out of scope for CoRL, but we don't believe the paper is out of scope and is not the reason for rejection.